# Molecular and spatial profiling of the paraventricular nucleus of the thalamus

Claire Gao[1,2]*, Chiraag A Gohel[3†], Yan Leng[1†], Jun Ma[1], David Goldman[3], Ariel J Levine[4], Mario A Penzo[1]*

[1]National Institute of Mental Health, Bethesda, United States; [2]Department of Neuroscience, Brown University, Providence, United States; [3]National Institute on Alcohol Abuse and Alcoholism, Rockville, United States; [4]National Institute of Child Health and Human Development, Bethesda, United States

**Abstract** The paraventricular nucleus of the thalamus (PVT) is known to regulate various cognitive and behavioral processes. However, while functional diversity among PVT circuits has often been linked to cellular differences, the molecular identity and spatial distribution of PVT cell types remain unclear. To address this gap, here we used single nucleus RNA sequencing (snRNA-seq) and identified five molecularly distinct PVT neuronal subtypes in the mouse brain. Additionally, multiplex fluorescent in situ hybridization of top marker genes revealed that PVT subtypes are organized by a combination of previously unidentified molecular gradients. Lastly, comparing our dataset with a recently published single-cell sequencing atlas of the thalamus yielded novel insight into the PVT's connectivity with the cortex, including unexpected innervation of auditory and visual areas. This comparison also revealed that our data contains a largely non-overlapping transcriptomic map of multiple midline thalamic nuclei. Collectively, our findings uncover previously unknown features of the molecular diversity and anatomical organization of the PVT and provide a valuable resource for future investigations.

*For correspondence:
claire.gao@nih.gov (CG);
mario.penzo@nih.gov (MAP)

[†]These authors contributed equally to this work

**Competing interest:** The authors declare that no competing interests exist.

## Editor's evaluation

This is an important study that identifies cell types within the paraventricular nucleus of the thalamus, a brain region with interesting functions. The conclusion that there are multiple cell types within this brain region that have distinct connectivity and transcriptomics is convincingly supported by the data provided. This work will be of interest to neurobiologists who seek to understand the cellular organization of the brain.

## Introduction

Recent models describe the PVT as a midline thalamic structure that integrates cortical, hypothalamic, and brainstem signals to drive adaptive behavioral strategies amid challenging situations (*Petrovich, 2021*; *Kirouac, 2021*; *McNally, 2021*; *Penzo and Gao, 2021*; *Kelley et al., 2005*). Consistent with this model, neuronal activity in the PVT is sensitive to both interoceptive and exteroceptive salient signals including hunger, reward, punishment, fear, and environmental cues that predict either positive and/or negative outcomes. In turn, the PVT plays a critical role in the signaling of emotional and motivational states largely via projections to the cortex, amygdala, and ventral striatum (*McGinty and Otis, 2020*; *Mátyás et al., 2018*; *Hua et al., 2018*; *Gao et al., 2020*; *Bhatnagar and Dallman, 1999*; *Stratford and Wirtshafter, 2013*; *Labouèbe et al., 2016*; *Sofia Beas et al., 2020*; *Meffre et al., 2019*; *Iglesias and Flagel, 2021*; *Li and Kirouac, 2008*; *Vertes and Hoover, 2008*; *Moga et al., 1995*; *Kirouac, 2015*). Given the PVT's involvement in such a diverse array of functions, there is growing

interest in determining how this structure is organized into functional subnetworks, particularly since broad manipulations of the PVT have often yielded disparate results (*Barson et al., 2020*; *McGinty and Otis, 2020*). For instance, while recent reports support the existence of a causal relationship between increased neuronal activity in the PVT and wakefulness, activation of a subpopulation of PVT neurons decreases wakefulness and promotes NREM sleep (*Mátyás et al., 2018*; *Hua et al., 2018*; *Gao et al., 2020*). Similar functional heterogeneity has been observed in the PVT's contributions to appetitive behaviors, with some studies reporting that lesions or pharmacological inactivation of the PVT increase food intake (*Bhatnagar and Dallman, 1999*; *Stratford and Wirtshafter, 2013*), while others show that direct or indirect activation of the PVT increases food-seeking behaviors (*Labouèbe et al., 2016*; *Sofia Beas et al., 2020*; *Meffre et al., 2019*).

Evidence suggests that the seemingly opposing roles of the PVT might be attributable to network differences arising from distinct cell-types that distribute along the rostro-caudal axis of the PVT (*Barson et al., 2020*; *Iglesias and Flagel, 2021*). Accordingly, the anterior and posterior subregions of the PVT (aPVT and pPVT, respectively) differentially innervate the amygdala, bed nucleus of the stria terminalis (BNST), hypothalamus, prefrontal cortex (PFC), ventral subiculum, and nucleus accumbens (NAc) and have been tied to different functions (*Li and Kirouac, 2008*; *Vertes and Hoover, 2008*; *Moga et al., 1995*; *Kirouac, 2015*). Although this broad classification into aPVT and pPVT subregions has aided advances in our understanding of the functional organization of the PVT, systematic dissections of the local organization of PVT subnetworks are currently lacking (*Barson et al., 2020*; *McGinty and Otis, 2020*; *Iglesias and Flagel, 2021*). Importantly, consistent with the notion that molecular identity can delineate cell types with different anatomical, functional, and electrophysiological properties (*Huang and Zeng, 2013*; *Huang, 2014*; *Fuzik et al., 2016*; *Oh et al., 2014*; *Luo et al., 2008*; *Luo et al., 2018*), studies have recently identified functional and anatomical differences that tie onto genetically defined subpopulations of PVT neurons (*Mátyás et al., 2018*; *Gao et al., 2020*; *Kessler et al., 2021*; *Engelke et al., 2021*). From this perspective, transcriptional profiling of the PVT could lead to valuable insights into the functional heterogeneity of this thalamic structure (*McGinty and Otis, 2020*).

In this study, we employ high throughput single-nucleus RNA sequencing and multiplex fluorescent in situ hybridization (ISH) labeling to identify five PVT subtypes that segregate differentially across the antero-posterior, medio-lateral, and dorso-ventral axes of the PVT. Our data highlights novel features about the molecular organization of the PVT, thereby offering a glimpse into how genetic diversity may tie into the various functions associated with this region of the thalamus. In addition, by performing the reciprocal principal component analysis (RPCA)-based integration and comparative analysis with the ThalamoSeq atlas, we find that PVT subtypes differentially innervate cortical areas and that our dataset contains complementary and previously unexplored mouse thalamic single-nuclei transcriptomes, including mediodorsal thalamus (MD) and intermediodorsal (IMD) thalamus (*Phillips et al., 2019*).

## Results

### Single-nucleus RNA-sequencing in and around the mouse PVT

To determine the unique transcriptional profiles of individual PVT neurons, single nuclei suspensions were first collected from tissue punches of the PVT and surrounding regions of the mouse brain and sequenced at the single-nuclei level (*Figure 1a*; See Methods; *Matson et al., 2018*). Next, low-quality nuclei and doublets were removed based on standard criteria (i.e. level of mitochondrial transcripts, number of genes), and integration using the Harmony algorithm was implemented to minimize experimental batch effects (*Figure 1—figure supplement 1*; *Korsunsky et al., 2019*). Clustering analyses were performed on 20,503 single-nuclei transcriptomes yielding a total of sixteen clusters visualized by uniform manifold approximation and projection (UMAP) (*Figure 1b and c*). The nuclei from these clusters were then assigned to seven major cell types based on their differential gene expression of canonical marker genes such as: astrocytes (*Agt*, *A2m*), endothelial cells (*Flt1*), ependymal cells (*Tmem212*), fibroblasts (*Cped1*), microglia (*Cx3cr1*), neurons (*Rbfox3*), oligodendrocyte precursor cells (OPCs) (*Pdgfra*), and oligodendrocytes (*Mal*) (*Figure 1d*, *Supplementary file 1*).

To assess neuronal cell types specifically, the eight clusters of neurons (1-8), containing a total of 9845 nuclei, were then filtered for nuclei containing greater than 1000 genes and re-clustered for

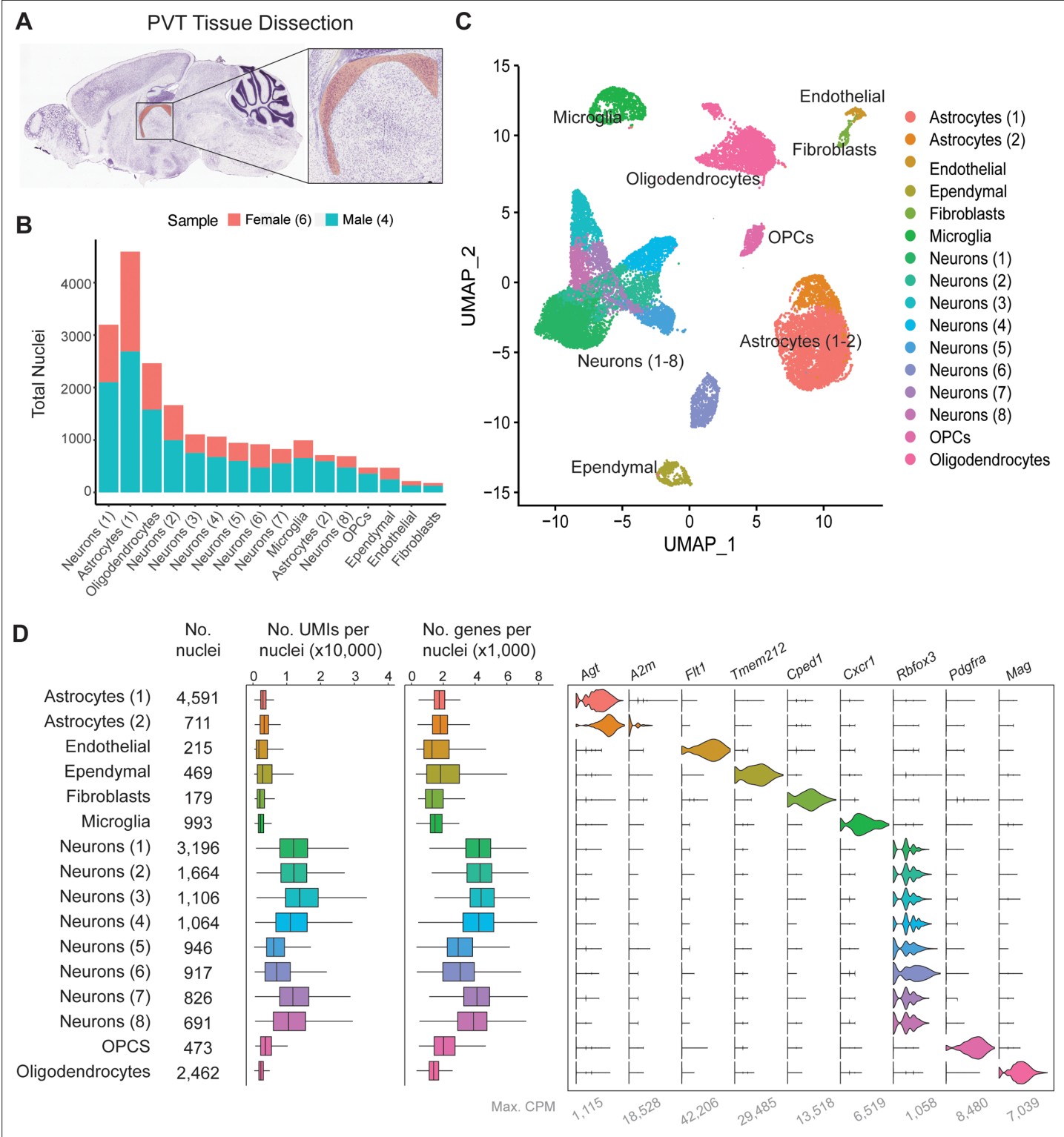

**Figure 1.** Single-nucleus RNA-sequencing in and around the mouse paraventricular nucleus of the thalamus (PVT). (**a**) Sagittal view of the PVT (red) illustrating the dissection target location. (**b**) Distribution of nuclei from four samples across all cell types. (**c**) The uniform manifold approximation and projection (UMAP) plot of all 13,220 nuclei from the combined dataset shows 14 cell clusters. (**d**) Cell type classification is based on the expression of marker genes in all 14 clusters. Left: box plot of UMI number in each cell cluster. Middle: box plot of genes detected per cell in each cell cluster. Right: violin plot showing expression profile of marker genes in 14 cell clusters. Max. CPM, maximum counts per million reads. Box plot legend: box is defined by 25th and 75th percentiles, whiskers are determined by 5th and 95th percentiles, and the mean is depicted by the square symbol.

*Figure 1 continued on next page*

*Figure 1 continued*

The online version of this article includes the following figure supplement(s) for figure 1:

**Figure supplement 1.** Quality control statistics and uniform manifold approximation and projection (UMAP) by sample.

**Figure supplement 2.** Clustering analysis and classification of all neurons.

further classification (*Figure 1—figure supplement 2a*). Clusters were characterized by referencing top gene markers from each cluster with their spatial expression from Allen Brain Atlas ISH data and previous literature (*Figure 1—figure supplement 2b, c*, https://mouse.brain-map.org/) (*Lein et al., 2007*). From this comparison, we found that PVT neurons (5737 nuclei; 58.3% of all neurons) expressed markers such as *Gck*, *C1ql3*, and *Hcn1*, and were thus represented in neuron clusters 0, 2, 3, and 4 (*Figure 1—figure supplement 2c, h*; *Gaspari et al., 2022*; *Chew et al., 2017*; *Kolaj et al., 2012*). Neurons associated with other brain regions surrounding the PVT and represented in our sample include: MD (1758 nuclei, 17.9% of neurons), IMD (609 nuclei, 6.19% of neurons), the principal nucleus of the posterior bed nucleus of the stria terminalis (BSTpr; 879 nuclei, 8.93% of neurons), and habenula (Hb; 862 nuclei, 8.76% of neurons) (*Figure 1—figure supplement 2d, g, h*, *Supplementary file 1*). Altogether, we observed nine unique neuronal clusters representing PVT and other adjacent nuclei, almost all of which are thalamic.

## Classification of PVT neuronal subtypes

To explore the cell type heterogeneity of neurons characteristic of the PVT, we removed all non-PVT neurons from subsequent analysis and focused our attention on the characterization of a total of 5737 PVT nuclei (see Methods). Clustering analysis of PVT neurons revealed five neuronal subtypes (PVT1-5), each expressing a unique genetic profile (*Figure 2a*). A phylogenetic tree was then constructed by generating a distance matrix between clusters in gene expression space. We found that PVT neurons were segregated into two major branches (*Figure 2b*). Specifically, PVT1, PVT2, and PVT5 subtypes were more closely related than PVT3, and PVT4 subtypes. Next, we performed differential gene expression analysis (DGEs) to select the top marker genes for each cluster (*Figure 2c–h*) (see Methods). Since our previous study identified two types of PVT neurons based on their expression of the *Drd2* gene or lack thereof, we compared the expression of *Drd2* across our five molecularly defined subtypes. The PVT1 (357 DGEs) subtype expressed *Drd2*, whereas PVT2 (184 DGEs) PVT3 (762 DGEs), PVT4 (295 DGEs) and PVT5 (440 DGEs) did not (*Figure 2—figure supplement 1*). These data suggest that, while as recently reported the PVT can be divided into Type 1 and Type 2 neurons based on *Drd2* expression, this view is oversimplified, and instead the PVT contains five molecularly distinct neuronal subtypes (*Gao et al., 2020*).

## Spatial distribution of five PVT neuron subtypes

We next validated the spatial distribution of our five putative PVT neuronal subtypes by employing multiplex ISH assays, where multiple rounds of ISH labeling and cleavage are performed in the same tissue sample. Using this method, we labeled the following cell type-specific markers: *Esr1* (PVT1), *Col12a1* (PVT2), *Npffr1* (PVT3), *Hcrtr1* (PVT4), and *Drd3* (PVT5) across the antero-posterior axis of the PVT (*Figure 3a–g*, *Supplementary file 1*). Following confocal imaging and post-hoc registration, we compared the relative expression of all five marker genes simultaneously across the anterior and posterior PVT. We observed that PVT1/*Esr1* and PVT2/*Col12a1* were significantly expressed in the pPVT, with little to no expression in the aPVT. In contrast, we found that PVT3/*Npffr1* and PVT4/*Hcrtr1* represented two aPVT biased cell types, which consistent with our previous report (*Gao et al., 2020*), are distributed in a gradient-like manner with low levels of expression localized to the dorsal pPVT. Finally, PVT5/*Drd3* was expressed across both the aPVT and pPVT in a pan-PVT-like manner (PVT5[pan]) (*Figure 3i–k*). We also performed quantitative analyses of our multiplex ISH-labeled samples to determine the distribution and number of positive cells for each transcript and generated a normalized transcript expression matrix (*Figure 3h*). From this matrix, we computed the Euclidian distances between gene markers and performed hierarchical clustering of our spatial data. Consistent with our sequencing results, we found that PVT3/*Npffr1* and PVT4/*Hcrtr1* expressing cells are more closely related than PVT1/*Esr1* and PVT2/*Col12a1* expressing cells (*Figure 3—figure supplement 1a*). Notably, hierarchical clustering based on our multiplex ISH data suggested that PVT3 and PVT4

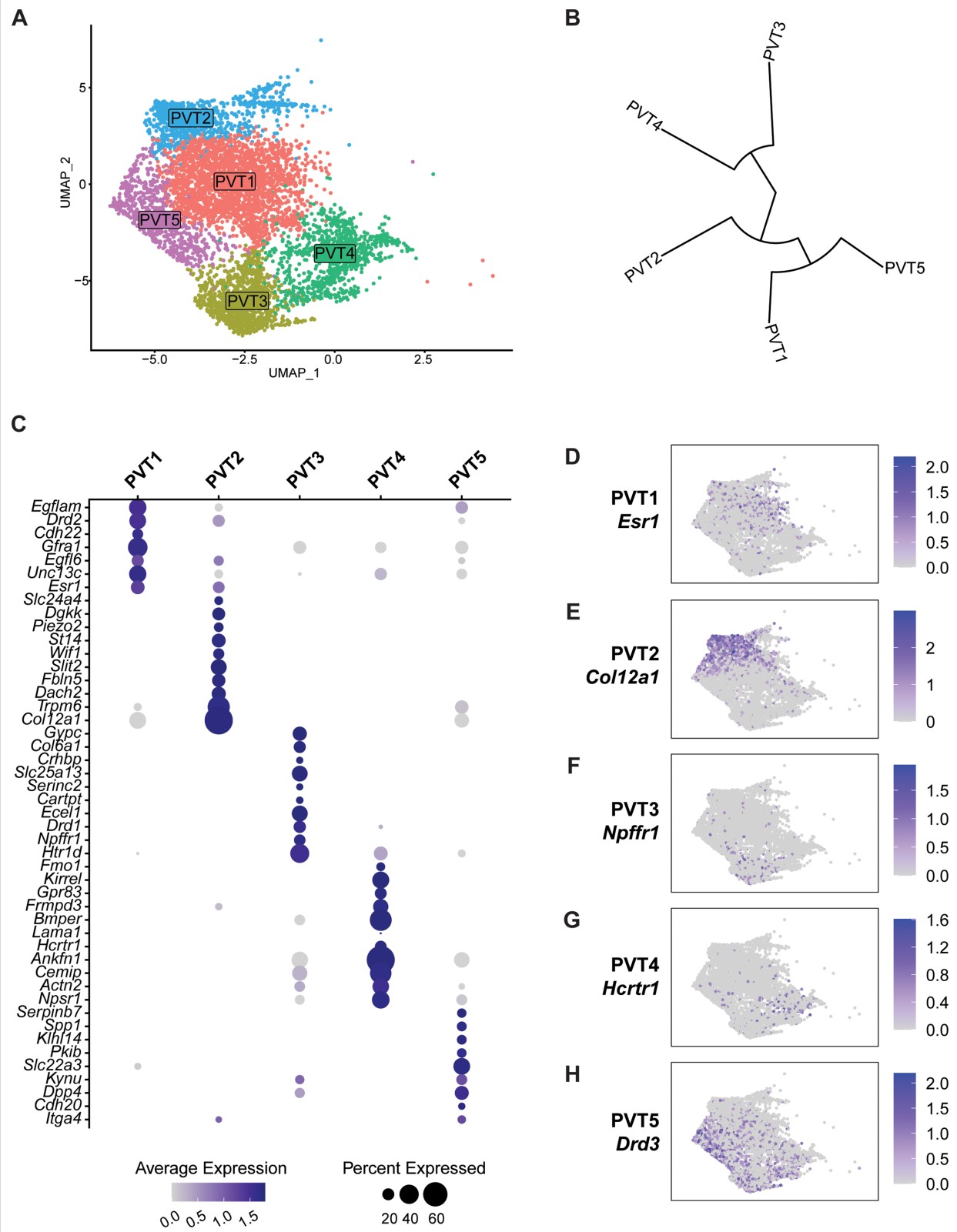

**Figure 2.** Five transcriptionally distinct neuronal subtypes are found in the paraventricular nucleus of the thalamus (PVT). (**a**) The uniform manifold approximation and projection (UMAP) plot of 5737 PVT neuronal nuclei shows five clusters. (**b**) Phylogenetic tree depicting cluster relationships based on the distance between clusters in gene expression space. (**c**) Dot plot of top gene marker average expression across PVT clusters selected based on pct. ratio value. (**d-h**) Feature plots of top marker genes (**d**) Esr1, (**e**) Col12a1, (**f**) Npffr1, (**g**) Hcrtr1, and (**h**) Drd3 for each PVT subtype.

*Figure 2 continued on next page*

*Figure 2 continued*

The online version of this article includes the following figure supplement(s) for figure 2:

**Figure supplement 1.** *Drd2* expression is found in PVT1 and PVT2.

are more closely related to PVT5 than PVT1 and PVT2. This difference from the sequencing phylogenetic tree is likely a result of how the distance matrices between the spatial data and the sequencing data are calculated. The spatial expression matrix compares one gene marker per cluster, providing an overview of the spatial relationship of each cluster, whereas the sequencing phylogenetic tree is generated using gene expression across all cells in a cluster, yielding a higher-resolution view of each cluster's transcriptional relationship (*Hao et al., 2021*). Together, these provide information on both the spatial context and transcriptional relationship of PVT cell types (*Ståhl et al., 2016*).

Plotting the locations of *Esr1*, *Col12a1*, *Npffr1*, *Hcrtr1*, and *Drd3* positive cells reveals a unique spatial segregation within the anterior-biased and posterior-biased clusters (*Figure 3i–j*). PVT4/*Hcrtr1* cells were localized more antero-medially (PVT4[AM]), whereas PVT3/*Npffr1* cells were expressed more antero-laterally (PVT3[AL]). Also, while PVT1/*Esr1* cells were located across the dorso-ventral axis of the pPVT (PVT1[P]), PVT2/*Col12a1* cells were restricted to the ventral portion of the pPVT and seemed to have the strongest expression at the bottom 'edge' of the pPVT (PVT2[edge]). To better visualize the spatial distribution among gene markers, we separately plotted pan-PVT (PVT5[pan]) cells with either the pPVT (PVT1[P], PVT2[edge]) or the aPVT (PVT3[AL], PVT4[AM]) biased clusters, and examined the percentage of overlap between positive cells amongst PVT subtypes (*Figure 3—figure supplement 1b–g*). Altogether, we find that, while there are varying degrees of overlap at the single-cell level, these gene markers largely represent spatially discrete cell types.

Importantly, while there are several markers that appear exclusive to PVT2[edge], some of the markers for PVT1[P] were also expressed to a weaker degree in PVT2[edge] (*Figure 2c*). Thus, to further confirm the nature of the spatial distribution of our five PVT subtypes, we selected a second set of markers for each of the aPVT biased (PVT3[AL], PVT4[AM]) and pPVT (PVT1[P], PVT2[edge]) biased subtypes and compared their spatial distribution to those of our prototypical PVT cluster markers (*Esr1*, *Col12a1*, *Npffr1*, and *Hcrtr1*) using multiplex ISH (*Supplementary file 1*). For this, we selected *Drd2*, *Pde3a*, *Insrr*, and *Npsr1*, as representative markers for subtypes PVT1[P], PVT2[edge], PVT3[AL], and PVT4[AM], respectively. Labeling of this second set of markers across the anterior and posterior PVT revealed a similar spatial distribution as our initial selected markers (i.e. AM, AL, P, edge) with only 30% or less overlap at the single-cell level among PVT subtype marker sets (*Figure 3—figure supplements 2 and 3*). These findings indicate that there is a high amount of diversity in gene expression even within the same PVT subtype and may suggest that, while there are only five molecular subtypes, cells within the same PVT subtype could have non-overlapping functions tied to differential gene expression.

Overall, in addition to validating the results of the snRNA-seq data, spatial mapping of our five PVT subtypes demonstrates the existence of spatially segregated, and molecularly distinct neuronal subpopulations of the PVT. As such, our data highlight a new feature of PVT cell type organization in which the PVT is structured by a combination of gradients: antero-posterior, dorsal-ventral, and medio-lateral. This arrangement of molecularly distinct cellular subtypes, wherein genetic identity and spatial distribution go hand in hand is a recurrent feature of the mammalian brain (*Phillips et al., 2019*; *Tasic et al., 2018*; *Cembrowski and Spruston, 2019*; *Romanov et al., 2019*; *O'Leary et al., 2020*; *Moffitt et al., 2018*). Importantly, molecular gradients in other thalamic nuclei, namely the thalamic reticular nucleus (TRN) and MD, have been linked to functional differences (*Li et al., 2020*; *Mukherjee et al., 2021*; *Martinez-Garcia et al., 2020*). The existence of a similarly close relationship between molecular identity and function among neurons of the PVT highlights how the discovery of additional molecular gradients may provide a framework for redefining the functional organization of the PVT (*Gao et al., 2020*; *Roy et al., 2022*).

## The five PVT subtypes have unique profiles of expression for neuromodulator receptors, neuromodulators, and ion channels

The PVT is a site of convergence of dense peptidergic innervation from cortical, hypothalamic, and hindbrain areas (*Penzo and Gao, 2021*; *Kirouac, 2015*; *Krout and Loewy, 2000b*; *Kirouac et al., 2005*; *Kirouac et al., 2006*; *García-Cabezas et al., 2007*; *Freedman and Cassell, 1994*; *Vertes*

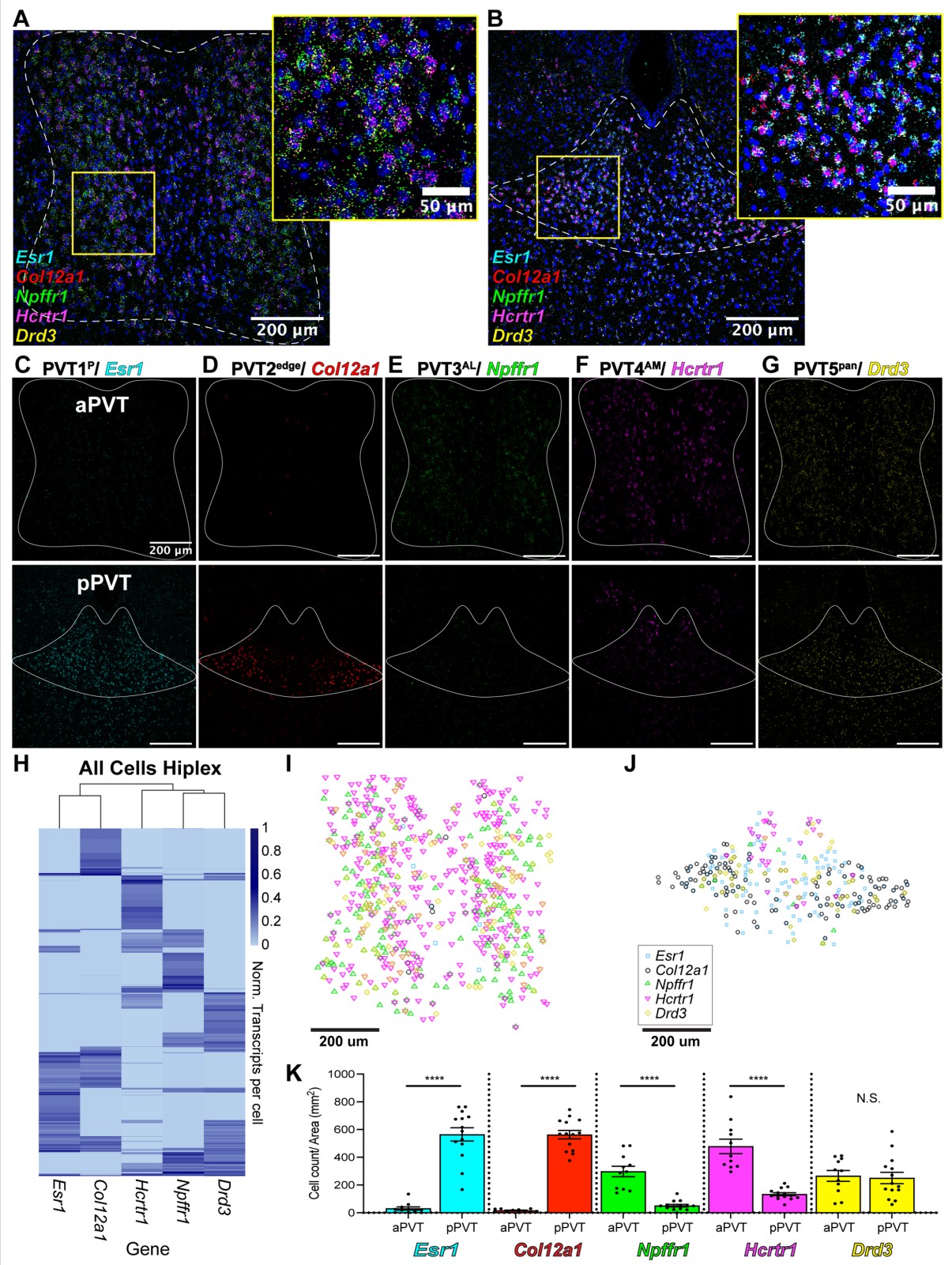

**Figure 3.** Mapping of five paraventricular nucleus of the thalamus (PVT) subtypes reveals a gradient-like segregated distribution. (**a**) RNA in situ hybridization in anterior PVT (white dotted outline) with closeup insert (right, yellow square) labeling one gene marker from each subtype: *Esr1* (light blue), *Col12a1* (red), *Npffr1* (green), *Hcrtr1* (magenta), and *Drd3* (yellow). (**b**) RNA in situ hybridization in posterior PVT (white dotted outline) with closeup insert (right, yellow square) labeling one gene marker from each subtype: *Esr1* (blue), *Col12a1* (red), *Npffr1* (green), *Hcrtr1* (magenta), and *Drd3* (yellow).

*Figure 3 continued on next page*

*Figure 3 continued*

(c–g) RNA in situ hybridization showing aPVT (top) and pPVT (bottom) of (c) *Esr1* from PVT1 subtype, (d) *Col12a1* from PVT2 subtype, (e) *Npffr1* from PVT3 subtype, (f) *Hcrtr1* from PVT4 subtype, (g) *Drd3* from PVT5 subtype. DAPI (blue). Scale bar 200 μm; all images are 20 X representative confocal images with brightness and contrast adjusted depicting expression patterns found in all sections from N=3 animals. (h) Heatmap of gene expression matrix with rows showing normalized transcripts per cell and columns showing gene markers from each PVT subtype; Left dendrogram displays hierarchical clustering by cell, top dendrogram displays hierarchical clustering by gene marker, and right bar shows heatmap legend. (i–j) Coordinates of positive cells from aPVT (i) and pPVT (j) with *Esr1* (blue square), *Col12a1* (black circle), *Npffr1* (green triangle), *Hcrtr1* (magenta downward triangle), *Drd3* (yellow circle) shown in the legend. (k) Bar graphs of the number of positive cells over the area (mm$^2$) per section of *Esr1* (blue; aPVT: 30.71±11.73; pPVT: 179.78±48.05; ****p=0.0000000015, two-sided Paired sample t-test), *Col12a1* (black; aPVT: 15.85±2.95; pPVT: 563.13±30.37; ****p=0.000000000000071, two-sided Paired sample t-test), *Npffr1* (green; aPVT: 297.64±37.93; pPVT: 51.03±9.13; ****p=0.00035), *Hcrtr* (magenta; aPVT: 478.53±52.05; pPVT: 133.62±10.76; ****p=0.00000021), and *Drd3* (yellow; aPVT: 265.97±38.52; pPVT: 250.43±41.18; N.S., not significant, p=0.79) in aPVT and pPVT. Data from N=3 animals are shown as mean ± SEM. Each data point represents one section.

The online version of this article includes the following figure supplement(s) for figure 3:

**Figure supplement 1.** Spatial distribution and overlap across anterior- or posterior- and pan-PVT subtypes.

**Figure supplement 2.** Overlap in gene expression among top markers from the same cluster.

**Figure supplement 3.** In situ hybridization of top markers from the same cluster.

---

*et al., 2010*; *Curtis et al., 2020*). As such, previous studies have indicated a segregated distribution and differential effects of neuromodulatory innervation and receptors along the antero-posterior axis of the PVT (*Pandey et al., 2019*; *Barson et al., 2015*; *Chen et al., 2015*; *Barson et al., 2017*; *Barrett et al., 2021*; *Matzeu et al., 2016*). To examine differences in genes that encode for proteins that regulate cellular excitability or function, we compared the expression of top differentially expressed neuromodulator receptors, neuromodulators, and ion channels across our five PVT subtypes (*Figure 4a–c*, *Supplementary file 1*). In addition to many neuromodulatory systems previously reported in the PVT, such as the orexin/hypocretin (*Hcrtr1/Hcrtr2*) (*Kirouac et al., 2005*), endogenous opioid (*Oprk1/Oprm1*) (*Chen et al., 2015*; *Vollmer et al., 2022*; *Zhu et al., 2016*; *Goedecke et al., 2019*; *Bengoetxea et al., 2020*), and dopamine system (*Drd2*) (*Clark et al., 2017*; *Beas et al., 2018*; *Zhang et al., 2015*; *Zhang et al., 2016*), our data revealed the existence of many previously uncharacterized neuromodulatory receptors and ion channel subunits across the five PVT subtypes. For example, *Npffr1*, the gene encoding neuropeptide FF receptor 1, is represented in the PVT3[AL] subtype, whereas *Ldlr*, the gene encoding for low-density lipoprotein receptor, is found in the PVT1 subtype (*Figure 4a*). Furthermore, within the same neuromodulatory system, we found differences in receptor family expression across PVT types. For instance, dopamine receptor genes *Drd1* and *Drd3* were identified in PVT3 [AL] and PVT5 [pan] subtypes, respectively, whereas *Drd2* predominated in the PVT1[P] subtype. Similarly, glycine receptor genes *Glra2* and *Glra3* were differentially represented across PVT4 and PVT5 subtypes, respectively (*Figure 4a*).

Given that differential expression of ion channel genes may be linked to divergence in the intrinsic membrane properties of PVT neurons, we compared ion channel gene expression across PVT types and observed substantial heterogeneity (*Figure 4b*; *Kolaj et al., 2012*; *Kolaj et al., 2014*; *Zhang et al., 2009*; *Zhang et al., 2010*). T-type calcium (Ca$^{2+}$) channels have been documented to contribute to the unique membrane properties and overall excitability of PVT neurons (*Kolaj et al., 2012*; *Kolaj et al., 2014*; *Zhang et al., 2009*; *Richter et al., 2005*). Interestingly, we found that while T-type Ca$^{2+}$ channel subunit gene *Cacana1h* was most robustly expressed in PVT3[AL] subtype, *Cacna1i* was differentially expressed in PVT2[edge] subtypes and to a lesser extent PVT5[pan] (*Figure 4b*). Similarly, we observed that potassium channel genes *Kcna1* and *Kcnk10* were differentially expressed in PVT3[AL] and PVT4[AM] subtypes. Such differential expression of voltage-gated potassium channels may further underlie variability in the membrane properties of PVT neurons (*Figure 4b*; *Kolaj et al., 2012*). Altogether, these findings highlight how our transcriptomic database may serve as a resource to identify key markers for future functional investigations, including genetic targeting of specific cell types.

Finally, we compared the gene expression of certain Ca$^{2+}$ binding proteins, including *Pvalb*, *Calb1*, and *Calb2* across the five PVT subtypes (*Figure 4d*). *Calb2*, the gene encoding calretinin, is a common gene marker used to identify PVT neurons that have recently been implicated in arousal signaling (*Mátyás et al., 2018*; *Hua et al., 2018*). Here, we find that *Calb2* expression is found in PVT1[P] and PVT5[pan], which is consistent with previous reports that expression of this gene is found across the PVT (*Mátyás et al., 2018*; *Hua et al., 2018*). *Calb1*, the gene encoding calbindin 1 (CB), and *Pvalb*,

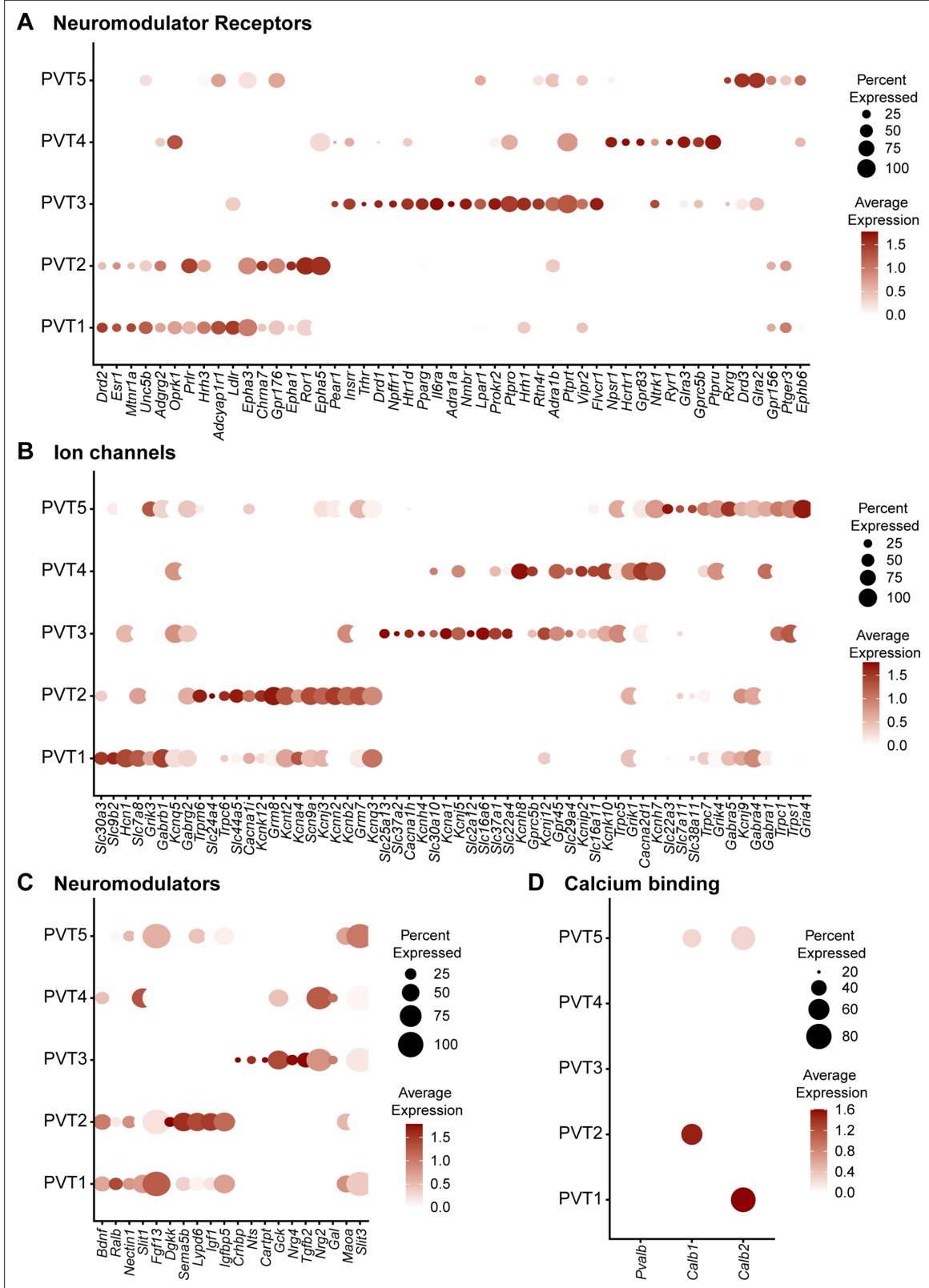

**Figure 4.** Five paraventricular nucleus of the thalamus (PVT) subtypes have diverse neuromodulator receptor, neuromodulator, and ion channel expression. (**a**) Dot plot depicting neuromodulator receptor gene expression across five PVT subtypes. (**b**) Dot plot depicting ion channel gene expression across five PVT subtypes. (**c**) Dot plot depicting neuromodulator gene expression across five PVT subtypes. (**d**) Dot plot depicting calcium binding gene (column) expression across five PVT subtypes. Legend: top, percent of nuclei in each cluster expressing a given gene; bottom, color intensity corresponding to average gene expression level.

the gene encoding parvalbumin (PV), are classical markers used in a thalamic classification system to segregate between what are known as 'core' and 'matrix' thalamic relay cells (*Jones and Hendry, 1989*). PV-expressing 'core' thalamic cells are defined by their topographical and dense projections to the middle layers of defined cortical regions, whereas CB-expressing 'matrix' thalamic cells instead send diffuse projections across unrestricted superficial cortical fields (*Jones, 1998a*; *Jones, 1998b*; *Jones, 2001*). From these projection patterns, 'core'-containing thalamic regions have often been deemed first-order thalamic nuclei, such as principal sensory thalamic relay nuclei (*Sherman and Guillery, 2002*). On the other hand, 'matrix'-containing regions that lack 'core' cells are often referred to as higher-order thalamic nuclei, such as the midline and intralaminar nuclei (*Sherman and Guillery, 2002*). Consistent with descriptions of the PVT as a higher-order midline thalamic structure, all five PVT subtypes lacked *Pvalb* expression (*Figure 4d*; *Jones and Hendry, 1989*; *Jones, 2001*). Notably, *Calb1* expression was only found in PVT2$^{edge}$ and PVT5$^{pan}$ subtypes, with higher expression in PVT2$^{edge}$. This result alongside other studies lends support to the notion that the core and matrix dichotomy is not a sufficient classification system to encompass all thalamic cell types (*Phillips et al., 2019*; *Clascá et al., 2012*; *Halassa and Sherman, 2019*).

## Cross-validation with ThalamoSeq dataset reveals an overlap between PVT subtypes and thalamic cortical projectors

To cross-reference our snRNA-seq dataset of the mouse midline thalamus, we performed integration using RPCA together with the recently published ThalamoSeq RNA-Seq atlas (thalamoseq.janelia. org), which contains single-cell sequenced thalamic cells that project to either motor, auditory, visual, somatosensory, or prefrontal cortices (PFC) (*Figure 5*, *Figure 5—figure supplement 1*; *Phillips et al., 2019*; *Hao et al., 2021*). Upon clustering analysis and plotting the UMAP of the integrated dataset with all midline-thalamic and BSTpr neurons, we found that our midline-thalamic dataset and the ThalamoSeq dataset contained mostly non-overlapping transcriptomes (*Figure 5—figure supplement 1*). Next, we performed integration of the ThalamoSeq dataset and PVT neurons specifically (*Figure 5a–b*). By mapping the integrated dataset based on cortical projection location from the ThalamoSeq cells or their original cluster identity (either PVT1-5 or ThalamoSeq), we could observe the cluster relationship between ThalamoSeq cortical projection location and our five PVT molecular subtypes (*Figure 5c–e*). DGE analysis on the integrated dataset revealed that our original PVT subtypes (PVT1-5) could be mapped onto clusters 0, 1, 3, 5, and 4, respectively, of the integrated dataset (*Supplementary file 1*, *Figure 5a and c*). Following this, comparing the proportion of cortical-projectors in the integrated dataset showed that the clusters representing PVT1$^P$ and PVT2$^{edge}$ contained overlap with PFC-projectors, whereas, unexpectedly, the cluster representing PVT4$^{AM}$ had a small amount of overlap with auditory and visual projectors (*Figure 5b*). The clusters representing PVT3$^{AL}$ and PVT5$^{pan}$ exhibited a small degree of overlap with prefrontal and visual cortical projectors, respectively (*Figure 5b*). Notably, the greatest amount of overlap was observed between ThalamoSeq PFC-projectors and the PVT2$^{edge}$ subtype, represented in cluster 1 of the integrated dataset. Next, we performed conserved marker analysis across these five integrated clusters and confirmed that *Col12a1* is a conserved marker across PFC-projectors and PVT2$^{edge}$ cells (*Figure 5f*, *Supplementary file 1*). These findings suggest that the *Col12a1*-positive PVT2$^{edge}$ subtype represents a subpopulation of neurons that project to the PFC, which is consistent with prior studies indicating that pPVT neurons are anatomically linked to the prelimbic cortex (PL) in a cortico-thalamocortical loop (*Gao et al., 2020*; *Li and Kirouac, 2008*; *Moga et al., 1995*; *Li and Kirouac, 2012*). To investigate whether, as predicted by our integration analysis PVT neurons projecting to the PFC reside along the ventral edge of the pPVT and express *Col12a1*, we injected green fluorescent retrobeads into the PL and used multiplex RNAScope to label *Col12a1* (PVT2$^{edge}$ marker) and *Drd2* (PVT1$^P$ marker) mRNA in posterior PVT sections. Consistent with the RPCA analysis, the results of this experiment revealed that the majority of retrobeads-labeled cells resided in the ventral edge of the PVT and partially co-localized with Col12a1-expressing cells (*Figure 5—figure supplement 2*). Also, in agreement with our results, anatomical tracing studies from the Mouse Connectome Project revealed that PVT efferent and afferent connectivity with the PL is restricted to the ventral edge of the pPVT (*Figure 5—figure supplement 3*, https://www.mouseconnectome.org/; *Zingg et al., 2014*). Given that the cluster representing PVT4$^{AM}$ overlapped with auditory and visual cortical projectors, we next employed the Allen Brain Mouse Connectivity Atlas Target Search Tool to look for auditory and visual areas that receive innervation by the PVT (*Figure 5—figure supplement*

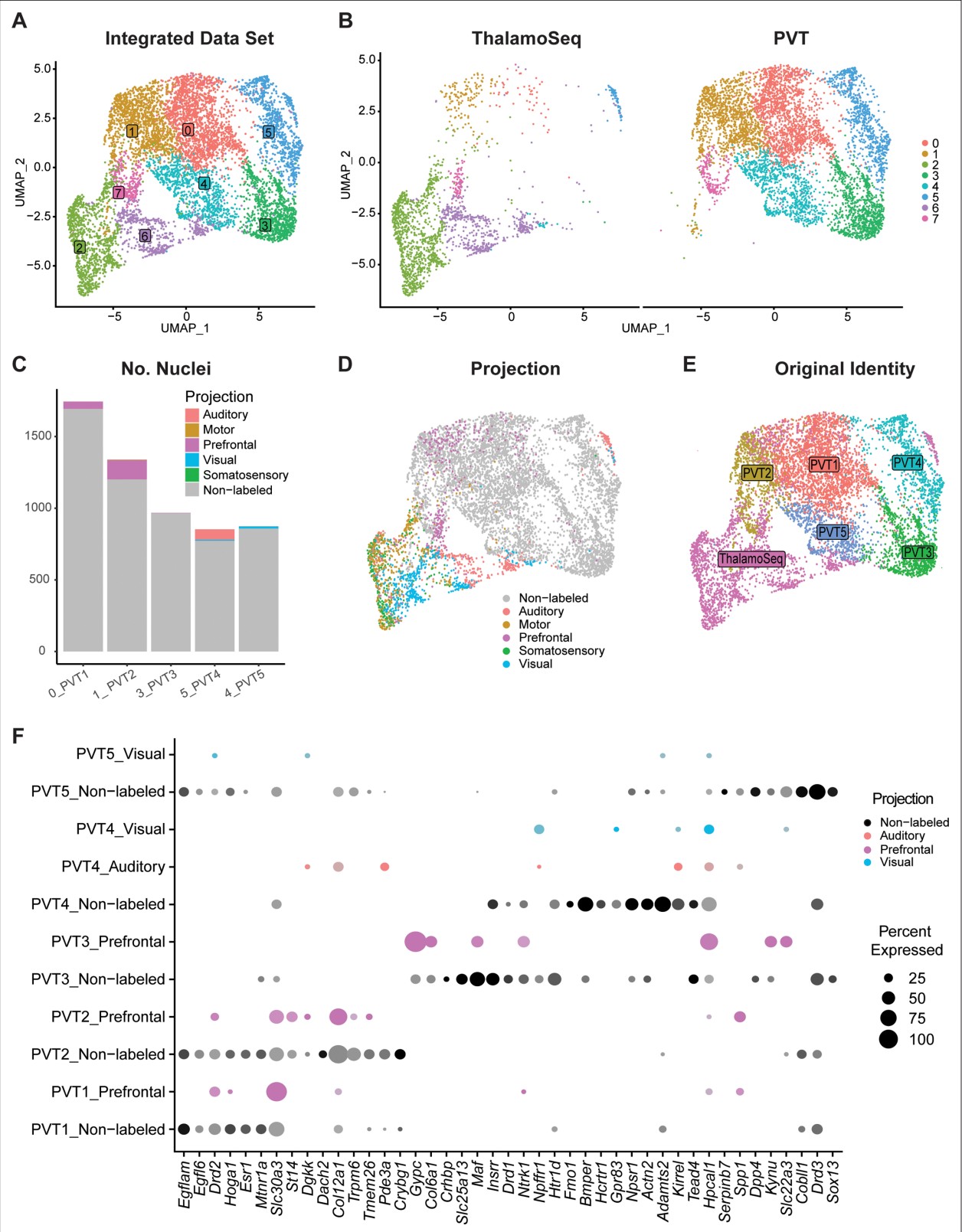

**Figure 5.** Cross-validation with ThalamoSeq dataset reveals overlap between paraventricular nucleus of the thalamus (PVT) subtypes and thalamic cortical projectors. (**a**) The uniform manifold approximation and projection (UMAP) plot of 7709 cells or nuclei from combined datasets annotated by cluster identity. (**b**) The same cells/nuclei are separated by the study of origin: ThalamoSeq (left) and present study (right) (**c**) Proportion of nuclei from each cortical projection target in the clusters that represent the molecular PVT subtypes from the integrated dataset: 0=PVT1, 1=PVT2, 3=PVT3,

*Figure 5 continued on next page*

*Figure 5 continued*

5=PVT4, 4=PVT5. (**d**) The same cells/nuclei are colored by the cortical projection target. (**e**) The same cells/nuclei are colored by the original identity. (**f**) Dot plot of PVT representing clusters (0,1,3, 5, 4) in the integrated dataset depicting expression of genes obtained from the conserved marker analysis (*Supplementary file 1*) split by projection identity.

The online version of this article includes the following figure supplement(s) for figure 5:

**Figure supplement 1.** RPCA-based integration with ThalamoSeq dataset and all neurons from the present study reveals mostly non-overlapping transcriptomes.

**Figure supplement 2.** PL-projecting pPVT neurons are localized to the ventroposterior edge of the paraventricular nucleus of the thalamus (PVT) and partially overlap with *Col12a1*.

**Figure supplement 3.** Dual anterograde and retrograde tracing from the prelimbic cortex (Mouse Connectome Project Experiment SW120125-02A).

**Figure supplement 4.** Projections from aPVT to Auditory and Visual cortical areas.

*4*, https://connectivity.brain-map.org/). Consistent with the results from our comparative analysis with the ThalamoSeq dataset, this search revealed four experiments demonstrating that anterior-biased thalamic regions including the aPVT (but not the pPVT) send projections to known auditory and visual cortical areas, such as the temporal association cortex (*Figure 5—figure supplement 4*). Collectively, the integrative analysis performed here supports that different PVT subtypes, namely PVT1$^P$, PVT2$^{edge}$, and PVT4$^{AM}$, share conserved markers with different thalamo-cortical projectors.

## Discussion

Despite being critical nodes in linking subcortical and cortical networks, to date, models of the spatial and molecular organization of midline and intralaminar thalamic nuclei have remained scarce (*Gao et al., 2020*; *Moga et al., 1995*; *Kirouac, 2015*; *Roy et al., 2022*; *Su and Bentivoglio, 1990*). Recently, our group and others have demonstrated that, contrary to the traditional discrete parcellations of thalamic nuclei, functionally distinct thalamic domains are organized in continuous neuronal gradients. Specifically, the PVT, TRN, and MD thalamus have each been shown to contain at least two genetically defined cell types that are organized in opposing spatial gradients and reflect distinct anatomical connectivity, electrophysiological signatures, and functional contributions (*Gao et al., 2020*; *Li et al., 2020*; *Mukherjee et al., 2021*; *Martinez-Garcia et al., 2020*). To expand this framework, in the present study, we generated an in-depth single-nuclei resolution transcriptomic atlas of the PVT that revealed the existence and spatial distribution of five previously undescribed neuronal subpopulations. Importantly, these PVT subtypes are organized in a combination of topographical gradients (e.g. medio-lateral, antero-posterior, dorso-ventral), which, to our knowledge, has not yet been reported in another midline thalamic area. However, given that thalamic pathways consist of repeated molecular architectures (*Phillips et al., 2019*), it is likely that this feature of multiple gradients is conserved and may serve as a basis for new conceptual models across the midline and intralaminar thalamic organization.

From our molecular and spatial mapping, we found that the four subtypes can be split into two major branches: (1) PVT1$^P$, PVT2$^{edge}$ and (2) PVT3$^{AL}$, PVT4$^{AM}$. These two branches consist of the posterior-biased, and anterior-biased subtypes, respectively, indicating that one major axis of separation between these two branches lies in their rostral-caudal distribution. While this notion is in partial agreement with classical anatomical studies delineating anterior and posterior PVT subregions, our study supports the ongoing revision of this terminology and fills a critical gap in our knowledge of the genetic heterogeneity of subpopulations within the PVT (*McGinty and Otis, 2020*). Notably, when selecting a second set of markers to label either anterior-biased or posterior-biased subtypes, we found that markers from the same cluster had a similar spatial distribution but labeled portions of non-overlapping neurons. This finding further serves to highlight the unexplored genetic heterogeneity of our PVT clusters and the functional implications of these intra- and inter-cluster differences should serve as a basis for future studies.

Overall, we report large variations in neuromodulator system and ion channel gene expression across PVT subtypes (*Figure 4c*), There is a formative body of evidence supporting that PVT neurons express unique intrinsic properties that undergo diurnal fluctuations associated with changes in gene expression (*Kolaj et al., 2012*). PVT neuronal firing properties are also documented to be affected

by a variety of neuromodulators, which may contribute to differences in their electrophysiological signatures (*Kolaj et al., 2012*; *Kolaj et al., 2014*). Additionally, neuromodulator influence in the PVT is higher than other neighboring thalamic regions, perhaps owing to its selective innervation by the hypothalamus and brainstem (*Kirouac, 2015*; *Krout and Loewy, 2000b*; *Krout et al., 2002*; *Vertes, 1991*; *Krout and Loewy, 2000a*). Given that each subtype contains the differential expression of neuromodulator receptor systems and ion channels, the previously reported effects of neuromodulators on the intrinsic properties of PVT neurons are likely to be diverse across subtypes (*Chen et al., 2015*; *Kolaj et al., 2014*; *Zhang et al., 2010*; *Zhang et al., 2006*; *Zhang et al., 2013*; *Kolaj et al., 2007*; *Ishibashi et al., 2005*; *Ong et al., 2017*; *Heilbronner et al., 2004*). To examine how PVT neurons undergo experience-dependent changes in gene expression, future investigations should include high throughput RNA-sequencing of the PVT following different biological conditions with the goal of comparing them with our atlas to identify potential biomarkers for therapeutic targeting. Indeed, based on the role of the rodent PVT in emotional processing and motivated behaviors, there is growing interest in the human PVT as a putative target for affective and substance abuse disorders, particularly since the human PVT has similar connectivity with the rodent PVT (*Kark et al., 2021*). Excitingly, the existence of some markers (e.g. *Drd2*, *Drd3*) reported in our present atlas has already been corroborated in the human PVT, supporting that our atlas may provide conserved gene markers across species (*Rieck et al., 2004*).

From classical tracing studies, the pPVT selectively innervates PL, nucleus accumbens core and ventromedial shell, central and basolateral amygdala, and lateral BST (*Li and Kirouac, 2008*; *Moga et al., 1995*). In line with this distinct connectivity, functional correlates implicate the pPVT as a key node in networks of aversive and reward-motivated behaviors, including stress and fear processing, and food- and drug-seeking. Importantly, our observation that there are two distinct posterior-biased subpopulations—PVT1$^P$ and PVT2$^{edge}$—gives rise to the question of whether these subtypes contribute to discrete functions. Of note, our integrative analysis with the ThalamoSeq atlas showed that PVT2$^{edge}$ cells in particular belong to a group of *Col12a1*-expressing PFC-projectors that reside on the ventroposterior edge of the PVT (*Phillips et al., 2019*). This result is consistent with classical tracing studies placing the pPVT in a reciprocal network with the PFC, and more specifically the PL (*Li and Kirouac, 2008*; *Moga et al., 1995*; *Li and Kirouac, 2012*). Intriguingly, anterograde viral tracing from multiple studies reveals that prelimbic afferents to the pPVT are spatially distributed in a manner that resembles the ventroposterior location of PVT2$^{edge}$ cells, lending support to the idea that PVT2$^{edge}$ cells may be preferentially integrated into a cortico-thalamocortical pathway that mediates its role in cognitive processes related to emotional and motivational states (*Figure 5—figure supplement 2*; *Zingg et al., 2014*; *Campus et al., 2019*; *Otis et al., 2019*). Indeed, prefrontal projections to the PVT guide the expression of emotional memory-associated behaviors such as conditioned fear and cue-reward food seeking (*Campus et al., 2019*; *Otis et al., 2019*; *Do-Monte et al., 2015*; *Quiñones-Laracuente et al., 2021*; *Otis et al., 2017*; *Haight and Flagel, 2014*). In this context, the PVT2$^{edge}$ cells may participate in the integration of cortical influence onto downstream subcortical targets to bias behavioral responding during learning and goal-oriented tasks where top-down modulation is necessary (*Campus et al., 2019*; *Otis et al., 2019*; *Do-Monte et al., 2015*; *Quiñones-Laracuente et al., 2021*; *Otis et al., 2017*; *Haight and Flagel, 2014*). In contrast, PVT1$^P$ neurons integrated to a lesser extent with PFC-projectors (*Figure 5c*). Interestingly, the medial portion of the pPVT, which lacks PVT2$^{edge}$ expression and is mostly populated by PVT1$^P$ neurons, is heavily innervated by subcortical inputs which may participate in biasing behavior toward stereotyped emotional reactions like those observed amid imminent environmental threats (*Cain and LeDoux, 2008*; *Martinez et al., 2013*; *Penzo et al., 2015*; *Li et al., 2014*; *Ma et al., 2021*). Additionally, PVT1$^P$ neurons specifically express *Esr1* (estrogen receptor 1), a cell marker for ventromedial hypothalamic (VMH) neurons that control aggression and mating, which are stereotyped and conserved behaviors under the control of subcortical networks (*Lee et al., 2014*; *Hashikawa et al., 2017*; *Falkner and Lin, 2014*; *Liu et al., 2022*). Whether *Esr1* neurons in the PVT contribute to the control of aggressive and mating behavior is not known; however, the PVT receives heavy afferents from *Esr1*-expressing VMH neurons, anatomically linking these two structures and prompting further investigation (*Lo et al., 2019*). Altogether, we propose that PVT2$^{edge}$ may consist of a genetic subpopulation that is preferentially involved in aspects of behavior requiring cortical processing, whereas the PVT1$^P$ subtype may contain neurons differentially recruited by tasks involving subcortical or cortical networks depending on their location.

It is important to highlight that, contrary to findings related to PVT2[edge] neurons and PVT1[P], RPCA-based integration with the ThalamoSeq dataset did not reveal additional major PFC-projectors among other PVT subtypes. One potential explanation for the lack of overlap between other PVT clusters and the ThalamoSeq atlas could lie in the retrograde labeling and sample collection method used (*Lähnemann et al., 2020*). Indeed, the authors of ThalamoSeq reported that their single cell collection was biased against midline thalamic nuclei due to issues related to sparse retrogradely labeled cells and thus, lack of overlap should not preclude other PVT subtypes, including PVT3[AL], as PFC-projectors (*Phillips et al., 2019*). One surprising observation generated by the RPCA-based integration of our dataset with the ThalamoSeq atlas is the notion that some neurons within PVT4[AM] may send anatomical projections to auditory and visual cortical areas (*Figure 5c and f*). This assertion is further supported by anterograde tracing studies showing that the aPVT projects to auditory and visual areas that have been implicated in or suggested to regulate limbic functions (*Figure 5—figure supplement 4*; *Dalmay et al., 2019*; *Cho et al., 2016*; *Yu et al., 2012*). Given the PVT's role in emotional salience processing, these findings raise the possibility that the PVT4[AM] participates in modulating emotional responses to salient stimuli in part through its interaction with limbic cortical areas beyond the PFC (*Gao et al., 2020*; *Haight et al., 2017*; *Zhu et al., 2018*; *Garau et al., 2022*).

In conclusion, while there have been some investigations into the molecular organization of the PVT, here we have provided the first comprehensive molecular and spatial profiling of cell types across the entirety of the PVT (*Phillips et al., 2019*; *Gaspari et al., 2022*; *Nagalski et al., 2016*). The PVT is a critical integrative node linking circuits of internal and external processing with those involved in arousal signaling, as well as instrumental and Pavlovian behavioral control (*Kirouac, 2021*; *McNally, 2021*; *Penzo and Gao, 2021*; *Kelley et al., 2005*). Our discovery of five spatially segregated PVT neuronal subtypes should serve as a basis for resolving previously conflicting observations and offer a starting point for new mechanistic inquiries as to how PVT offers control over its diverse range of functional correlates.

## Methods

### Mice

All procedures were performed in accordance with the *Guide for the Care and Use of Laboratory Animals* and were approved by the National Institute of Mental Health (NIMH) Animal Care and Use Committee. Mice used in this study were housed under a 12 hr light-dark cycle (6 a.m. to 6 p.m. light), with food and water available ad libitum. The following mouse lines were used: C57BL/6 J (The Jackson Laboratory). Male and female mice 8–12 weeks of age were used for all experiments.

### Single-nuclei RNA-sequencing

#### Sample collection and single nucleus RNA sequencing

Tissue punches from the entire antero-posterior PVT from four P90 adult C57BL/6 J mice (12 weeks) were dissected and pooled together for each individual sample and a total of four biological replicates were collected. Nuclei were obtained using the mechanical-detergent lysis protocol described step-by-step in *Matson et al., 2018*. Following sample preparation, the single nuclei suspensions were delivered to collaborators at the NHLBI genomics core facility and processed for single-cell sequencing using the 10 X Genomics Chromium Single Cell 3' Kit. Samples were sequenced using the Illumina, Inc NovaSeq 6000 system to yield a single nuclei dataset. 10 x Genomics Cellranger 3.0.0 was used to map sequences to a reference mouse genome (*Zheng et al., 2017*).

#### Clustering analysis

The data were analyzed using the R package Seurat version 4.0 developed by the Satija Lab (*Hao et al., 2021*). Clustering was performed in three phases on (1) all cell types, (2) all neurons, and (3) putative PVT neurons, based on the methods described in *Russ et al., 2021* with the following adaptations (*Russ et al., 2021*). For all three phases, each cluster was analyzed for candidate marker genes and excluded if the cluster met either of the following criteria. Clusters were considered low-quality if they had fewer than three significant markers relevant to cell type, particularly if they showed very low nGene (<100). Clusters were considered doublets if they had significant markers for multiple unrelated

cell types and boxplots indicate they had a significant range of nGene. Differential gene analysis was performed using the Wilcoxon Rank Sum test with log fold change >0.5 and p value adjusted <0.05.

For phase one, data from all samples were pooled together and filtered based on the following criteria: nuclei containing less than 200 genes (to filter empty droplets), nuclei with greater than 5% mitochondrial genes were removed (to filter low-quality nuclei). The dataset was normalized using the SCTransform package and the highest variable genes were used to perform principal component analysis (PCA). The Harmony algorithm was used to penalize clusters biased by sample origin (*Korsunsky et al., 2019*). The most significant principal components were determined by selecting the minimum threshold based on: (1) the point at which the PCs cumulatively represented 90% of the standard deviation and (2) the point at which the percent variation between consecutive PCs is less than 0.1%. This value was then manually compared to the PC elbow plot and inspection of the contributing gene lists. 18 PCs were used for clustering. To select cluster resolution, a range of values was tested from 0.1 to 0.8, and UMAPs and top differentially expressed gene lists were generated and visually inspected, and resolution 0.7 was selected. Nuclei were clustered and visualized by using UMAP. Cell types were classified using DropViz and based on the presence of well-established marker genes (http://dropviz.org) (*Saunders et al., 2018*). For phase 2, raw data from all cells in neuronal clusters were used, filtered for nuclei containing at least 1000 detected genes, re-scaled, re-normalized, and re-integrated. The top six PCs were selected and resolution 0.3 was selected, using the approach described above. Here, neurons were then classified based on (1) a comparison of cluster markers with known markers and with known co-expression patterns in the literature or Allen Brain In Situ Hybridization Atlas (http://mouse.brain-map.org). For phase 3, targeted sub-clustering was performed to investigate PVT-specific subtypes. The above procedure was repeated and 18 PCs and a resolution of 0.4 were used. Clustering resolution was selected by calculating the highest mean silhouette score across cluster resolutions (*Patterson-Cross et al., 2021*). Top marker genes were selected for each PVT cluster by calculating the ratio of expression for a particular gene across the active cluster (pct. 1) and all other clusters (pct. 2) and selecting for the highest ratio value (pct. ratio) (Supp. File 1 c). Higher pct. ratio values indicate markers with higher specificity to a given cluster.

## Stereotaxic surgery

Stereotaxic methods for retrobeads injections were performed using previously described procedures (*Penzo et al., 2015*) and an AngleTwo stereotaxic device (Leica Biosystems) at the following stereotaxic coordinates: PL,+1.85 mm from Bregma, –0.55 mm lateral from midline, and –2.30 mm vertical from the cortical surface. Following all surgical procedures, animals recovered on a heating pad and returned to their home cages after 24 hr post-surgery recovery and monitoring. Animals received subcutaneous injections with Metacam (meloxicam, 1–2 mg/kg) for analgesia and anti-inflammatory purposes. Green fluorescent retrobeads (LumaFluor, Inc) were injected at a total volume of approximately 0.2 µl and were allowed seven days for maximal expression.

## In situ hybridization
### Sample preparation and ISH procedure (RNAscope)

Fresh-frozen brains from adult male C57BL/6 J mice (8–12 weeks) were sectioned at a thickness of 16 µm using a Cryostat (Leica Biosystems). Sections were collected onto Superfrost Plus slides (Daigger Scientific, Inc), immediately placed on dry ice, and subsequently transferred to a –80 °C freezer. *Esr1*, *Col12a1*, *Insrr*, *Hcrtr1*, *Drd2*, *Npffr1*, *Npsr1*, *Pde3a* mRNA signal was labeled by using the Hiplex RNAscope Fluorescent kit v2 (Advanced Cell Diagnostics), according to manufacturer's instructions. Sections were cover slipped using Diamond Prolong antifade mounting medium with DAPI (ThermoFisher Scientific).

### Signal detection and analysis

After the amplification procedure, slides were examined on a Nikon A1R HD confocal microscope (Nikon) using a 20 X objective. Images were first processed by removing background noise using the background subtraction tool (5.0-pixel rolling ball radius) in FIJI (Image J). Images from the same section were then registered using RNAscope Hiplex Image Registration Software v2. The signal was subsequently quantified with CellProfiler using a freely available pipeline (macros) for RNAscope (*Patterson-Cross et al., 2021*). A protocol with a step-by-step description of how to implement

this pipeline for analyzing RNAscope data was recently published (*Erben et al., 2018*; *Erben and Buonanno, 2019*). This pipeline was modified to include up to 12 mRNA probes and cells were considered positive for a given gene if they contained a minimum of seven mRNA transcripts (dots). Subsequently, all RNAscope data was analyzed by a blind experimenter. Sections from bregma −0.22 to −0.34 were considered anterior PVT and sections from bregma −1.58 to −1.70 were considered posterior PVT. Gene expression matrices were generated and analyzed using RStudio Version 4.2.0. Heatmaps and hierarchical clustering were performed using R functions pheatmap() and hclust().

## Merged analysis and integration

### Published data acquisition

Published data from the ThalamoSeq project were downloaded from the NCBI Gene Expression Omnibus public functional genomics data repository. Single-cell data and metadata from *Phillips et al., 2019* (GSE133911) were downloaded as processed count matrices (https://www.ncbi.nlm.nih.gov/geo/query/acc.cgi?acc=GSE133911) (*Phillips et al., 2019*).

### RPCA-based integration and clustering analysis

Count matrices for each dataset were merged to obtain the full data file. Uniform data filtering was applied across the merged file. All cells and nuclei with at least 1000 detected genes (to exclude low-quality neurons) and less than 5% of transcripts being mitochondrial (to exclude lysing cells or mitochondria-nuclei doublets) were analyzed, yielding a total of 8507 nuclei.

The merged data were analyzed using Seurat version 4.0 (*Hao et al., 2021*). The integration was performed using Seurat version 4.0 Standard Workflow (RPCA) Integration such that data are first normalized with SCTransform (method = 'glmGamPoi') and PCA is run individually on each dataset prior to integration. Integration anchors were calculated using 30 PCs and used to integrate the data.

The most significant principal components were identified by elbow plot and manual inspection of the contributing gene lists and 13 PCs were used for clustering. To select cluster resolution, a range of values was tested from 0.2 to 0.6, and UMAPs and top differentially expressed gene lists were generated, and resolution 0.35 was selected. Conserved markers for integrated data cluster 1 were obtained using FindConservedMarkers() function to confirm that these PVT/Prefrontal-projectors express both *Col12a1*.

## Statistics and data presentation

All data were imported to OriginPro 2016 (OriginLab Corp.) for statistical analyses. Initially, normality tests (D'Agostino-Pearson and Kolmogorov-Smirnov) were performed to determine the appropriateness of the statistical tests used. All data are presented as mean ± SEM. No assumptions or corrections were made prior to data analysis. Differences between the two groups were always examined using a two-sided Student's t-test, where $p < 0.05$ was considered significant and $p > 0.05$ was considered non-significant. The sample sizes used in our study are about the same or exceed those estimated by power analysis (power = 0.8, $\alpha = 0.05$). Graphs were generated in OriginPro or Rstudio and figures were generated using Adobe Illustrator. For RNAscope experiments, the sample size is 2–3 mice. All experiments were replicated at least once, and all subjects were age-matched.

## Acknowledgements

We thank the NHLBI Genomics Core for performing library preparation and sequencing our samples. We thank Drs. Sofia Beas and Hugo Tejeda for their comments on the manuscript. We thank Drs. Ruchi Komal, Maria Yurgel, and Anton Schulmann for their training and guidance. This work was supported by the NIMH Intramural Research Program (1ZIAMH002950) and (in part) by the Division of Intramural Research of the NIH, NINDS (ZIANS003153). The content is solely the responsibility of the author(s) and does not necessarily represent the official views of the National Institutes of Health.

## Additional information

### Funding

| Funder | Grant reference number | Author |
|---|---|---|
| National Institute of Mental Health | 1ZIAMH002950 | Mario A Penzo |
| National Institute of Neurological Disorders and Stroke | ZIANS003153 | Ariel J Levine |

The funders had no role in study design, data collection and interpretation, or the decision to submit the work for publication.

### Author contributions

Claire Gao, Conceptualization, Data curation, Formal analysis, Validation, Investigation, Visualization, Methodology, Writing – original draft, Writing – review and editing; Chiraag A Gohel, Software, Formal analysis, Validation, Visualization; Yan Leng, Data curation, Validation, Visualization, Methodology; Jun Ma, Data curation, Methodology; David Goldman, Supervision, Writing – review and editing; Ariel J Levine, Supervision, Funding acquisition, Methodology, Writing – review and editing; Mario A Penzo, Conceptualization, Resources, Data curation, Supervision, Funding acquisition, Validation, Writing – original draft, Project administration, Writing – review and editing

### Author ORCIDs

Ariel J Levine http://orcid.org/0000-0002-0335-0730
Mario A Penzo http://orcid.org/0000-0002-5368-1802

### Ethics

All procedures were performed in accordance with the Guide for the Care and Use of Laboratory Animals and were approved by the National Institute of Mental Health Animal Care and Use Committee (see Methods - Mice).

### Decision letter and Author response

Decision letter https://doi.org/10.7554/eLife.81818.sa1
Author response https://doi.org/10.7554/eLife.81818.sa2

## Additional files

### Supplementary files

• MDAR checklist

• Supplementary file 1. Lists of differential gene expression for snRNA-seq data and conserved marker analysis from the integrated dataset.

### Data availability

All RNA-seq data generated in our study have been deposited into the Gene Expression Omnibus repository (GSE208707). Raw images of RNAscope experiments are publicly available at: https://figshare.com/s/e2918829cabfdd0392fb.

The following datasets were generated:

| Author(s) | Year | Dataset title | Dataset URL | Database and Identifier |
|---|---|---|---|---|
| Gao C, Gohel CA, Leng Y, Goldman D, Levine AJ, Penzo MA | 2022 | Molecular and spatial profiling of the paraventricular nucleus of the thalamus | https://www.ncbi.nlm.nih.gov/geo/query/acc.cgi?acc=GSE208707 | NCBI Gene Expression Omnibus, GSE208707 |
| Gao C, Gohel CA, Leng Y, Goldman D, Levine AJ, Penzo MA | 2023 | Molecular and spatial profiling of the paraventricular nucleus of the thalamus | https://figshare.com/s/e2918829cabfdd0392fb | figshare, e2918829cabfdd0392fb |

The following previously published dataset was used:

| Author(s) | Year | Dataset title | Dataset URL | Database and Identifier |
|---|---|---|---|---|
| Schulmann A, Phillips JW, Hara E, Wang L, Lemire AL, Nelson SB, Hantman AW | 2019 | Transcriptomic atlas of thalamic nuclei | https://www.ncbi.nlm.nih.gov/geo/query/acc.cgi?acc=GSE133911 | NCBI Gene Expression Omnibus, GSE133911 |

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
