## [Editor Report]

This is an important study that identifies cell types within the paraventricular nucleus of the thalamus, a brain region with interesting functions. The conclusion that there are multiple cell types within this brain region that have distinct connectivity and transcriptomics is convincingly supported by the data provided. This work will be of interest to neurobiologists who seek to understand the cellular organization of the brain.

---

## [Decision Letter]

**Decision letter after peer review:**

Thank you for submitting your article "Molecular and spatial profiling of the paraventricular nucleus of the thalamus" for consideration by *eLife*. Your article has been reviewed by 3 peer reviewers, and the evaluation has been overseen by a Reviewing Editor and Laura Colgin as the Senior Editor. The following individual involved in review of your submission has agreed to reveal their identity: James M Otis (Reviewer #2).

Essential revisions:

All of the reviewers felt that the work fills an important gap in understanding of how the PVT can have distinct functions and that the data will be valuable for future studies in the field. We feel that one piece of data needs to be added and there are some minor corrections to the paper.

1) Two of the reviewers questioned why only male mice were used in the study – no justification was given for this approach. More importantly, given the evidence cited by the authors that ESR1 functions in some of the cells within the PVT, all of the reviewers felt upon discussion of this point that comparing single cell gene expression in the same brain region from female mice is a key test for interpreting the functional relevance of the transcriptomic differences in PVT cell populations, and thus is within the scope of the current study. In addition, these data will provide a replication experiment to verify the rigor of the classification. This is the only new piece of data or data analysis we feel is essential for the current study.

2) There are minor points identified by each of the reviewers below with respect to some text descriptions, figure presentations, and typos for the authors to correct.

*Reviewer #1 (Recommendations for the authors):*

1. The authors used only male mice. They note that at least one of the cells of interest expresses ESR1. Why did they use only males?

2. Pg. 17 "Importantly, these PVT subtypes are organized in a combination of topographical gradients (e.g., medio-lateral, antero-posterior, dorso-ventral), which, to our knowledge, has not yet been reported in another midline thalamic area." Indeed this may be the first time these gradients have been observed in this very specific brain region, but gradients have been reported for example in amygdala and are likely to be common once people start confirming seemingly digital sequencing data with analog in situ quantification.

3. The paper is fine for being rather short – it is interesting data – however the discussion is much too long compared with the volume of original data and should be cut to a more reasonable size. Currently for a 300 line study (results start at line 80) the discussion is 200 lines.

*Reviewer #2 (Recommendations for the authors):*

Moderate Concerns

1. Several of the heat maps use white to represent average gene expression or low expression, which blends into the white background of the figure. While this makes the enriched gene sets stand out, it also makes it impossible to visualize the percent expression (e.g., see Figure 2C). The authors could consider including a different heatmap color scheme, without white, or the authors could outline each dot with a thin black line. In addition to being more descriptive, this could prevent the misleading conclusion that certain genes are not expressed in the presented cell type.

2. While the characterization of 5 PVT cell types is interesting, I do think that the authors could dial back the conclusions that there are exactly 5 PVT cell types. Based on the cell clustering (e.g., Figure 2a), the degree of separation between these cell types is not obvious. In other words, if these circles were drawn in black, I would have no idea where each cell type separates itself from another. This highlights a point that the authors themselves make within the manuscript, that there could be heterogeneous functions within these cell types. I think that this point is someone focused and understated, as there are likely other ways to classify cell types based on gene expression that would not result in these specific 5 groups.

3. Related to the above, using alternative clustering methodologies would provide additional information about the replicability, or bias, of these groups of neurons using the current hierarchical clustering scheme. The authors should use one or two alternative clustering algorithms to determine the reproducibility of their data.

4. Additionally, the authors should provide silhouette plots to show how well each cell 'fits' into the isolated clusters.

*Reviewer #3 (Recommendations for the authors):*

I think this dataset will be of broad interest to the scientific community, and it is the first to focus specifically on the PVT. However, the data seems preliminary given the low number of confirmed PVT cells, and that it was conducted in male mice only. I would recommend increasing the size of the dataset by running equal number of female mice to hopefully double the number of PVT neurons. This would also allow for examining sex differences in the PVT across cell types, which would be an added value. While this is a substantial ask, I do think it could be completed within a few months (since these are just naïve mice). I also think this will greatly assist in selecting the most appropriate clustering resolution for the data as mentioned above.

There are many places related to the identification of cell type markers, that the authors rely on the Allen Brain Atlas for gene expression data to confirm their results. I would caution the authors from doing this. The in situ hybridization data in the Allen Brain Atlas likely contains many false negatives (genes not being accurately detected) because it was done prior to the develop of HCR/RNAscope which are much more sensitive.

---

## [Author Response]

Essential revisions:All of the reviewers felt that the work fills an important gap in understanding of how the PVT can have distinct functions and that the data will be valuable for future studies in the field. We feel that one piece of data needs to be added and there are some minor corrections to the paper.1) Two of the reviewers questioned why only male mice were used in the study – no justification was given for this approach. More importantly, given the evidence cited by the authors that ESR1 functions in some of the cells within the PVT, all of the reviewers felt upon discussion of this point that comparing single cell gene expression in the same brain region from female mice is a key test for interpreting the functional relevance of the transcriptomic differences in PVT cell populations, and thus is within the scope of the current study. In addition, these data will provide a replication experiment to verify the rigor of the classification. This is the only new piece of data or data analysis we feel is essential for the current study.

We thank the reviewers for this point and agree that female samples are an important addition for providing a resource on the transcriptomic cell types of the PVT. As such, in the revised manuscript we have added 7,501 nuclei collected from adult female mice to our original dataset of 13,002 nuclei collected from adult male mice, for a total of 20,503 nuclei. Combined analyses of these additional samples confirm the original findings of 5 PVT cell types and their respective markers.

Reviewer #1 (Recommendations for the authors):1. The authors used only male mice. They note that at least one of the cells of interest expresses ESR1. Why did they use only males?

We thank the reviewer for raising this concern and have added female mice to the study. With the addition of female mice (as noted above under Essential Revisions), we found that each cell type contains a similar number of nuclei from both males and females and *Esr1* still emerges as a top marker for PVT1 cell type. In Author response image 1 are feature plots for *Esr1* split by sex showing the presence of *Esr1* in both the male and female samples. We agree that the reviewer raises an important point regarding previous literature on differences in estrogen receptor profile across sex. While we did not examine sex differences in the present study, we hope that our dataset can provide a resource for future studies to examine more closely the presence of sex-related differences in gene expression in the PVT.

**Author response image 1. sa2fig1:** 

2. Pg. 17 "Importantly, these PVT subtypes are organized in a combination of topographical gradients (e.g., medio-lateral, antero-posterior, dorso-ventral), which, to our knowledge, has not yet been reported in another midline thalamic area." Indeed this may be the first time these gradients have been observed in this very specific brain region, but gradients have been reported for example in amygdala and are likely to be common once people start confirming seemingly digital sequencing data with analog in situ quantification.

We agree with the reviewer that gradients have previously been reported in other areas and have highlighted this in the revised manuscript (Lines 190-202).

3. The paper is fine for being rather short – it is interesting data – however the discussion is much too long compared with the volume of original data and should be cut to a more reasonable size. Currently for a 300 line study (results start at line 80) the discussion is 200 lines.

We thank the reviewer for this suggestion and have shortened the discussion accordingly.

Reviewer #2 (Recommendations for the authors):Moderate Concerns1. Several of the heat maps use white to represent average gene expression or low expression, which blends into the white background of the figure. While this makes the enriched gene sets stand out, it also makes it impossible to visualize the percent expression (e.g., see Figure 2C). The authors could consider including a different heatmap color scheme, without white, or the authors could outline each dot with a thin black line. In addition to being more descriptive, this could prevent the misleading conclusion that certain genes are not expressed in the presented cell type.

We thank the reviewer for their suggestion. We have adjusted the color scheme in Figure 2C to be a continuous color without white for better visualization of the markers.

2. While the characterization of 5 PVT cell types is interesting, I do think that the authors could dial back the conclusions that there are exactly 5 PVT cell types. Based on the cell clustering (e.g., Figure 2a), the degree of separation between these cell types is not obvious. In other words, if these circles were drawn in black, I would have no idea where each cell type separates itself from another. This highlights a point that the authors themselves make within the manuscript, that there could be heterogeneous functions within these cell types. I think that this point is someone focused and understated, as there are likely other ways to classify cell types based on gene expression that would not result in these specific 5 groups.

We thank the reviewer for raising this point. We have added a portion of the discussion to further highlight the gradient like manner and heterogeneity of our neuronal subtypes (Lines 340-346).

3. Related to the above, using alternative clustering methodologies would provide additional information about the replicability, or bias, of these groups of neurons using the current hierarchical clustering scheme. The authors should use one or two alternative clustering algorithms to determine the reproducibility of their data.

To cluster our nuclei, we utilize the Seurat v3 package which first uses principal component analysis to perform a linear dimensionality reduction of the single nuclei data. To determine the number of principal components (PCs), we select the minimum threshold based on: (1) the point at which the PCs cumulatively represent 90% of the standard deviation and (2) the point at which the percent variation between consecutive PCs is less than 0.1%. Following PC selection, we implement Seurat’s graph-based clustering approach which clusters cells based on their Euclidian distance in PCA space using a Knearest neighbor graph. This graph-based clustering approach has been commonly implemented in multiple studies (Macosko et al., 2015, *Cell*; Xu & Su, 2015, *Bioinformatics*; Levine et al., 2015, *Cell*).

Following the reviewer’s recommendation, we have now calculated the similarity between each pair of clusters from our 5 PVT cell types using the *pairwiseModularity* function from the bluster package. This function computes a modularity score which is defined as the difference between the observed and expected edge weights between clusters based on a null model of random connections between nodes

(https://rdrr.io/github/LTLA/bluster/man/pairwiseModularity.html). From this we see the highest modularity scores corresponding to the same cluster and lower modularity scores between other clusters, which further serves to support that our 5 PVT cell types represent transcriptionally unique clusters. In Author response image 2, we also see that PVT1 has a lower on diagonal modularity score which is consistent with what we see upon spatial validation of markers from this subtype, and which has also been noted in the manuscript.

4. Additionally, the authors should provide silhouette plots to show how well each cell 'fits' into the isolated clusters.

We thank the reviewer for their suggestion. Accordingly, we have now calculated the mean silhouette score across clusters at different clustering resolutions. In Author response image 3 we select the resolution with the highest mean silhouette score as the cluster resolution for our PVT cell types.

**Author response image 3. sa2fig3:**